# Regulating the expression of gene drives is key to increasing their invasive potential and the mitigation of resistance

Andrew Hammond[1,2], Xenia Karlsson[1], Ioanna Morianou[1], Kyros Kyrou[1], Andrea Beaghton[1], Matthew Gribble[1], Nace Kranjc[1], Roberto Galizi[1], Austin Burt[1], Andrea Crisanti[1,3]*, Tony Nolan[4]*

1 Department of Life Sciences, Imperial College London, London, United Kingdom, 2 Department of Molecular Microbiology and Immunology, Bloomberg School of Public Health, Johns Hopkins University, Baltimore, Maryland, United States of America, 3 University of Padova, Padova, Italy, 4 Liverpool School of Tropical Medicine, Liverpool, United Kingdom

* acrs@imperial.ac.uk (AC); tony.nolan@lstmed.ac.uk (TN)

**Data Availability Statement:** All raw data used for the assembly of all figures is included as supplementary excel files. Raw sequencing reads (amplicon sequencing) from experiments

## Abstract

Homing-based gene drives use a germline source of nuclease to copy themselves at specific target sites in a genome and bias their inheritance. Such gene drives can be designed to spread and deliberately suppress populations of malaria mosquitoes by impairing female fertility. However, strong unintended fitness costs of the drive and a propensity to generate resistant mutations can limit a gene drive's potential to spread.

Alternative germline regulatory sequences in the drive element confer improved fecundity of carrier individuals and reduced propensity for target site resistance. This is explained by reduced rates of end-joining repair of DNA breaks from parentally deposited nuclease in the embryo, which can produce heritable mutations that reduce gene drive penetrance.

We tracked the generation and selection of resistant mutations over the course of a gene drive invasion of a population. Improved gene drives show faster invasion dynamics, increased suppressive effect and later onset of target site resistance. Our results show that regulation of nuclease expression is as important as the choice of target site when developing a robust homing-based gene drive for population suppression.

## Author summary

Gene drives are selfish genetic elements that are able to drastically bias their own inheritance. They can rapidly invade populations, even starting from a very low frequency.

Recent advances have allowed the engineering of gene drives deliberately designed to spread genetic traits of choice into populations of malaria-transmitting mosquito species–for example traits that impair a mosquito's ability to reproduce or its ability to transmit parasites.

The class of gene drive in question uses a very precise cutting and copying mechanism, termed 'homing', that allows it to increase its numbers in the cells that go on to form

described in this project is deposited at the European Nucleotide Archive under Project ID: PRJEB37873.

**Funding:** This work was supported by grants from the Bill & Melinda Gates Foundation. The funders had no role in study design, data collection and analysis, decision to publish, or preparation of the manuscript.

**Competing interests:** The authors have declared that no competing interests exist.

sperm or eggs, thereby increasing the chances that a copy of the gene drive is transmitted to offspring.

However, while this type of gene drive can rapidly invade a mosquito population, mosquitoes can also eventually become resistant to the gene drive in some cases. Here we show that restricting the cutting activity of the gene drive to the germline tissue is crucial to maintaining its potency and we illustrate how failure to restrict this activity can lead to the generation of mutations that can make mosquitoes resistant to the gene drive.

## Introduction

### Gene drives and malaria control

Gene drives are genetic elements that are capable of biasing their own inheritance, allowing their autonomous spread in a population, even from a very low initial frequency. Coupling a trait of interest to a drive element is a way of deliberately modifying a population, potentially in a very short timeframe. In the case of the mosquito vector of malaria, gene drives have been proposed to spread traits that either interfere with the mosquito's capacity to reproduce, or its capacity to transmit the malaria parasite.

We know that vector control is effective in controlling malaria–the malaria burden was halved in the period 2000–2015 and the vast majority of this gain was achieved through targeting the vector with conventional insecticide-based approaches (bednets and residual spraying) [1]. However, resistance to insecticides is stalling these gains [2]. Gene drive is a technology with the potential to augment and complement existing control approaches in a self-sustaining way.

### Endonuclease-based homing gene drives

Gene drives based on site-specific endonucleases were first proposed over 15 years ago [3] and recent advances in CRISPR technology have led to several demonstrations that this endonuclease, which is easy to reprogram to recognise a genomic site of choice, can be repurposed as a gene drive [4,5].

The premise is that the endonuclease is sufficiently specific to recognise a DNA target sequence within a region of interest and the gene encoding the endonuclease is inserted within this target sequence on the chromosome, thereby rendering it immune to further cleavage. When a chromosome containing the endonuclease is paired with a chromosome containing the wild-type target site, the site is cleaved to create a double stranded break (DSB) that can be repaired, either through simple 'cut and shut' non-homologous end-joining (NHEJ) or through homology-directed repair (HDR). HDR involves strand invasion from the broken strand into regions of immediate homology on the intact chromosome, and synthesis across the intervening region to repair the gap. In the arrangement described this can lead to copying of the endonuclease, and its associated allele, from one chromosome to another in a process referred to as 'homing'. If homing takes place in the germline then inheritance of the gene drive is biased because the majority of sperm or eggs produced in the germline will inherit the drive, on either the original gene drive-carrying chromosome or the newly converted chromosome. The rate of spread of this type of gene drive is thus a product of the rate of germline nuclease activity and the probability with which double stranded breaks are repaired by the host cell using HDR rather than end-joining repair.

## Previous limitations of CRISPR-based homing gene drives

We and others have tested several gene drives designed to suppress or modify mosquito populations [6–8]. With any gene drive the force of selection for resistance to the drive will be proportional to the fitness cost imposed by the drive. In the case of population suppression approaches, this selection falls on the mosquito. We previously built several CRISPR-based gene drives designed to achieve population suppression through disrupting female fertility genes and demonstrated their spread throughout caged populations of the malaria mosquito, *Anopheles gambiae* [7]. However, the drive was eventually replaced in the population by resistant mutations generated by end-joining repair in the fraction of cleaved chromosomes that were not modified by homing [9]. To ultimately replace the drive in a population, these mutations must also encode a functional copy of the target gene so that they restore fertility to females.

In determining the propensity for target site resistance to arise, the amount of functional constraint at the target site determines the degree of variation that can be tolerated there. In the first iteration of a population suppression gene drive that we built, the target site sequence is poorly conserved [7], suggesting little functional constraint, and therefore is particularly prone to accommodating resistant alleles that restore function to the target gene ('r1' alleles, as opposed to 'r2' alleles which are resistant to cleavage but do not restore function to the target gene) (S1 Fig) [9,10]. The importance of choosing functionally constrained sites was shown when a gene drive targeting a highly conserved target sequence in the female-specific isoform of a sex determination gene was able to crash caged populations without selecting for resistant mutations of the r1 class [8]. Notwithstanding this success, it is unlikely that any single site will be completely 'resistance-proof'. This is true for any suppressive technology, from antibiotics to insecticides, and measures to prolong the durability of these interventions need to be considered at inception.

In addition to multiplexing gene drives to recognise more than one target site [3,4,11,12], akin to combination therapy with antibiotics, it is necessary to reduce the relative contribution of the error-prone end-joining repair pathway, over HDR, since this can serve to increase the range and complexity of potential resistant alleles on which selection could act.

Cleavage by maternally deposited nuclease in the embryo is believed to be the major source of end-joining mutations in current gene drive designs [6,9,13]. We have previously observed high levels of end-joining repair in the embryo following strong maternal deposition using the *vas2* promoter. This is likely a consequence of germline expression that persists through oocyte development leading to perduring Cas9 transcript and/or protein in the newly fertilised embryo. Additionally, using the same promoter, unrestricted activity of the endonuclease outside of the mosquito germline caused a strong unintended fitness cost in females harbouring a single copy of the gene drive, due to partial conversion of the soma to the homozygosity for a null allele. These fitness effects retarded the invasive potential of the drive for two reasons: 1- reduced bias in inheritance each generation due to low fecundity of drive-carrying (heterozygous) females; 2- increased selection pressure for resistant mutations [9].

We tested a suite of germline promoters for their ability to restrict nuclease activity to the germline and their ability to limit the generation of resistance. We employed genetic screens to get a quantitative determination of the range of mutations arising when gene drive constructs are driven by the different promoters as well as where and how these mutations were generated.

One of the promoters characterised here has previously been used in a gene drive construct that was able to successfully eliminate a laboratory population of mosquitoes [8]. Importantly, we quantify here the magnitude of the improvement in gene drive performance afforded by choice of this promoter and the resulting improved regulation of expression alone, rather than

choice of gene drive target. We then employed the best performing constructs in a cage invasion assay to determine how our single generation estimates of homing performance translated to our mathematical models of penetrance and population suppression over time, while tracking the emergence of resistant alleles under selection.

Our results show that the regulation of nuclease expression is a key aspect in ensuring the robustness and efficacy of homing-based gene drive systems for population control.

## Results and discussion

### Choice of germline promoters

To find alternatives to the *vas2* promoter, we investigated three mosquito genes, *AGAP006241* (*zero population growth*, *zpg*), *AGAP006098* (*nanos*, *nos*) and *AGAP007365* (*exuperantia*, *exu*), that are expressed in the germline of male and female *Anopheles gambiae* [14] and may show reduced somatic expression or deposition into the embryo.

In *Drosophila* the gene *zpg* is specifically expressed in the male and female germline, where it mediates the formation of gap junctions between the developing germline and cyst cells [15]. The mosquito ortholog of *zpg* shows a similar expression pattern [14] and appears to be functionally conserved as it is essential for both male and female gonad development [16]. *Exu* and *nos* are maternal effect genes in *Drosophila* that are transcribed in the oocyte and deposited into the early embryo [17–19]. Crucially, deposited *nos* and *zpg* mRNA concentrate at the germ plasm due to regulatory signals present on the untranslated regions, which also further restrict translation of maternal mRNAs to the germline [15,20]. The promoter region of *exuperantia* has been validated in *Drosophila* [21] and in the tiger mosquito, *Aedes aegypti*, has been used to drive robust expression in both male and female germlines and has recently been used to control expression of Cas9 in a split gene drive system in this mosquito [22,23]. In contrast to *zpg* and *exu*, several reports have suggested that *nos* is specific to the female germline in mosquitoes, however promoter fusions in *Aedes aegypti* and *Anopheles gambiae* led to low level expression in males perhaps due to incomplete recapitulation of endogenous gene expression [24–26].

### Generation of new gene drives designed to restrict spatiotemporal expression to the mosquito germline

To determine the effect of transcriptional control of gene drive activity, we tested the new constructs at a previously validated female fertility locus that was prone to resistance [9], in order to quantify any reduction in the creation or selection of resistant mutations that resulted from reducing embryonic end-joining and improving female fertility, respectively.

The target site in question resides within the gene *AGAP007280* (Fig 1A), an ortholog of the *Drosophila* gene *nudel* required in the follicle cells to define polarity of the eggshell [27,28]. Null mutations at this target site cause a recessive female sterile phenotype. The active gene drive cassette (*CRISPR^h*) used previously to target *AGAP007280* [7] was modified to contain Cas9 under control regulatory sequences flanking upstream and downstream of either *zpg*, *nos* or *exu* genes. The *CRISPR^h* constructs were inserted within the *AGAP007280* target site, by recombinase-mediated cassette exchange, as previously [7] (Fig 1B). The new gene drive strains were named *nos-CRISPR^h*, *zpg-CRISPR^h*, *exu-CRISPR^h*, respectively.

### *Zpg* and *nos* promoters drive high levels of homing in the germline and vary in magnitude of maternal effect

Germline gene drive activity that leads to homing is expected to cause super-Mendelian inheritance of the drive. However, maternally-deposited Cas9 has the potential to cause resistant

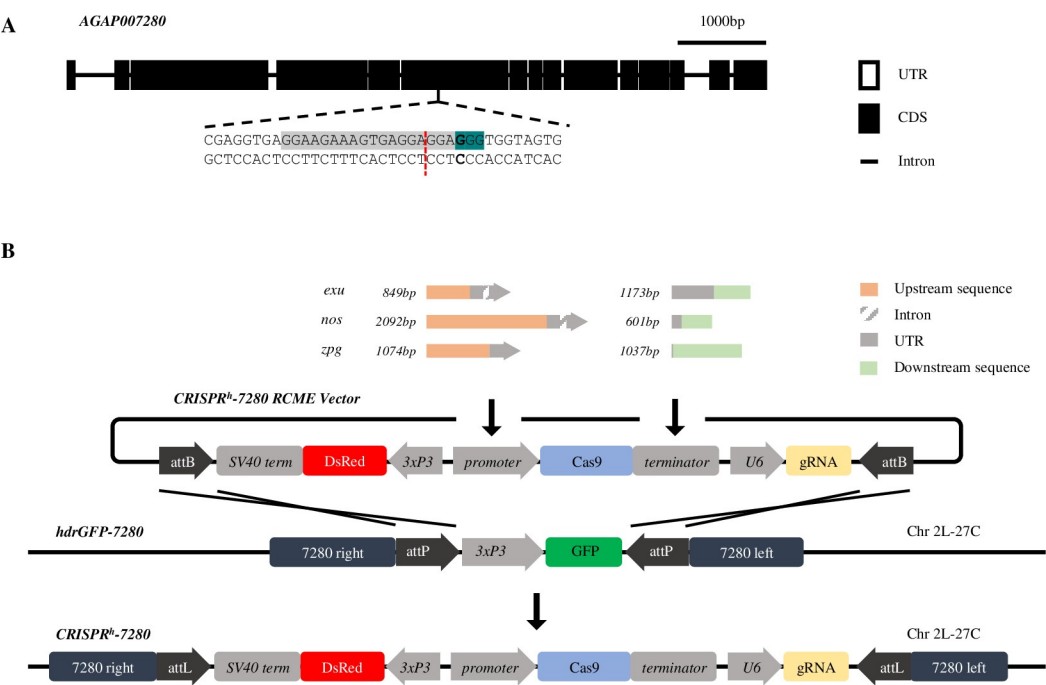

**Fig 1. Target site and design of new *CRISPR^h* gene drives designed to express Cas9 under regulation of *zpg, nos* and *exu* germline promoters.** (A) The haplosufficient female fertility gene, *AGAP007280*, and its target site in exon 6 (highlighted in grey), showing the protospacer-adjacent motif (highlighted in teal) and cleavage site (red dashed line). (B) *CRISPR^h* alleles were inserted at the target in *AGAP007280* using φC31-recombinase mediated cassette exchange (RMCE). Each *CRISPR^h* RMCE vector was designed to contain *Cas9* under transcriptional control of the *nos*, *zpg* or *exu* germline promoter and terminator, a gRNA targeted to *AGAP007280* under the control of the ubiquitous *U6* PolIII promoter, and a *3xP3*::*DsRed* marker.

mutations in the embryo that may reduce the rate of homing during gamete formation (if the mutations occur in germline precursor cells) or reduce fertility (due to somatic mosaicism of null mutations in the target fertility gene) [6,9,10,13,29].

Assays were performed to measure the transmission rates and fertility costs associated with each of three drives (Fig 2). Individuals heterozygous for the gene drive were crossed to wild type and their progeny scored for the presence of the DsRed marker gene linked to the construct. To test the magnitude of any parental effect, gene drive carriers were separated according to whether they received their gene drive allele paternally or maternally. Data from our previously generated gene drive constructs under control of the *vasa* promoter (*vas2-CRISPR^h*) served as a benchmark [7].

When the gene drive was received paternally *zpg-CRISPR^h* transmission rates of 93.5% (±1.5% s.e.m.) in males and 97.8% (±0.6% s.e.m.) in females were observed, falling only slightly below the *vas2-CRISPR^h* rates in the equivalent cross (99.6% ±0.3% s.e.m. in males and 97.7% ±1.6% s.e.m. in females) (Fig 2D and 2B). The paternally-received *nos-CRISPR^h* construct showed comparable rates to *vas2-CRISPR^h*, with 99.6% (±0.3% s.e.m.) inheritance in females and 99.1% (±0.3% s.e.m.) inheritance in males (Fig 2F). In contrast, paternally-received *exu-CRISPR^h* showed only modest homing rates in males (65.0% ±2.0% s.e.m. transmission rate) and no homing in females (Fig 2H).

When the gene drive was received maternally, the *vas2-CRISPR^h* constructs showed a large reduction in homing in males (60.2% ±1.9% s.e.m. inheritance rate, equivalent to homing of 20.4% of non-drive chromosomes) compared to those that received the drive paternally

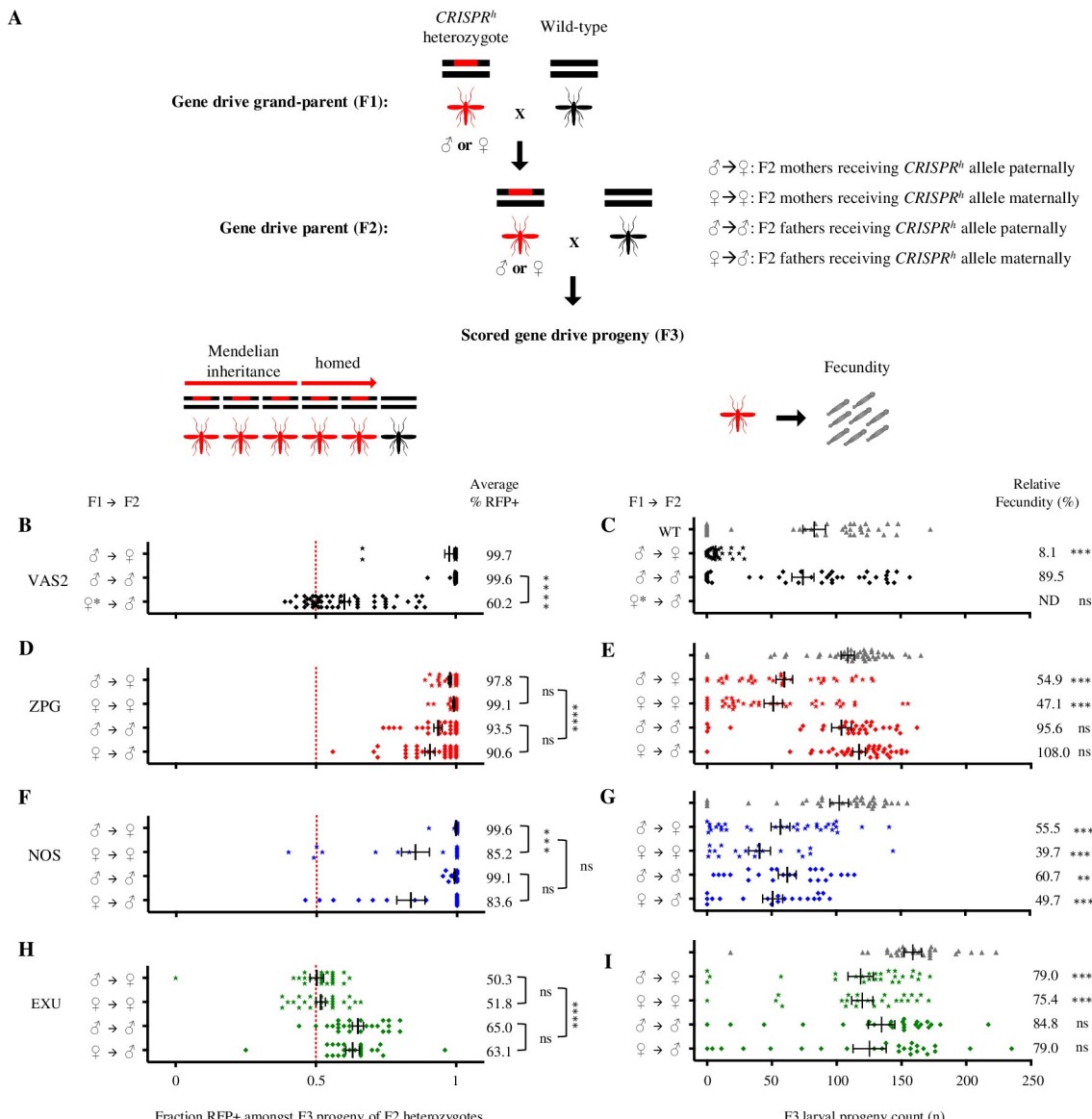

**Fig 2. Comparison of fecundity and homing rates in mosquitoes containing Cas9-based gene drives under regulation of *zpg*, *nos* and *exu* germline promoters.** Phenotypic assays were performed to measure fecundity and transmission rates for each of three drives. *CRISPR^h* heterozygotes of both sexes were crossed to wild type (panel A, F1) and their *CRISPR^h* heterozygous progeny were subsequently also crossed to wild type (A, F2). The larval progeny (F3) of the F2 cross were counted to determine the fecundity of *CRISPR^h* heterozygotes (F2), depending on whether they inherited the gene drive from male or female parents (F1) (A right, C, E, G, I). The larval progeny (F3) were also scored for the presence of DsRed linked to the construct to determine gene drive transmission rates of *CRISPR^h* heterozygotes (F2), depending on whether they inherited the gene drive from male or female parents (F1) (A left, B, D, F, H). Each data point represents one clutch from one individual F2 female. Average transmission rates are shown on the right of panels B, D, F, H (Mann-Whitney test, ****: $p < 0.0001$, ***: $p < 0.001$, *: $p < 0.05$, ns: non-significant), and relative fecundity to wild type is depicted on the right of panels C, E, G, I (Kruskal-Wallis test, ****: $p < 0.0001$, ***: $p < 0.001$, **: $p < 0.01$, ns: non-significant). Males and females were further separated by whether they had inherited the *CRISPR^h* construct from either a male or female parent (Mann-Whitney test, ****: $p < 0.0001$, ***: $p < 0.001$, *: $p < 0.05$, ns: non-significant). For example, ♂→♀ denotes the progeny count and gene drive transmission rates in the progeny of a heterozygous *CRISPR^h* female that had inherited the drive allele from a heterozygous *CRISPR^h* male, when crossed to a wild-type male. High levels of homing were observed in the germline of *zpg-CRISPR^h* and *nos-CRISPR^h* males and females, however the *exu* promoter generated only moderate levels of homing in the germline of males but not females (Mann-Whitney test, p < 0.0001). The significant maternal effect upon homing performance in offspring previously seen for *vas2-CRISPR^h* (Hammond et al., 2017, reproduced here in panel B) was also observed in *nos-CRISPR^h*, but not for *zpg-CRISPR^h* (Mann-Whitney test, ****: $p < 0.0001$, **: $p < 0.001$, *: $p < 0.05$, ns: non-significant). Counts of hatched larvae for the individual crosses revealed improvements in the fertility of heterozygous females containing *CRISPR^h* alleles based upon *zpg*, *nos* and *exu* promoters compared to the *vas2* promoter. Phenotypic assays were performed on G2 and G3 for *zpg*, G3 and G4 for *nos*, and G15

for *exu*. ♀* denotes *vas2-CRISPR*<sup>h</sup> females that were heterozygous for a resistance (r1) allele, these were used because heterozygous *vas2-CRISPR*<sup>h</sup> females are usually sterile. ND = not determined.

(99.2% homing of non-drive chromosomes) (Fig 2B) (Mann-Whitney test, p<0.0001). On the contrary, for the *zpg-CRISPR*<sup>h</sup> construct, we saw minimum transmission rates of 90.6% (±1.8% s.e.m.) in males and a higher rate of 99.1% in females (±0.4% s.e.m.) (Mann-Whitney test, p = 0.0019), yet no significant maternal or paternal effect (p>0.9999) (Fig 2D). With the *nos-CRISPR*<sup>h</sup> construct, though homing rates were still high, we saw a maternal effect leading to reduction in homing performance in females (85.2% ±5.0% s.e.m. vs 99.6% ±0.3% s.e.m., respectively, Mann-Whitney test, p = 0.0006) and a similar but non-significant reduction in males (83.6% ±5.0% s.e.m. inheritance when inherited maternally vs 99.1% ±0.4% s.e.m. when paternally inherited, Mann-Whitney test, p = 0.066) (Fig 2F). This suggests that Cas9 from the *nos-CRISPR*<sup>h</sup> construct is maternally deposited, though to a lesser extent than the *vas2-CRISPR*<sup>h</sup> constructs, and active in a way that leads to the formation of alleles in the zygotic germline that are resistant to homing in the next generation. Finally, maternally-inherited *exu-CRISPR*<sup>h</sup> constructs showed modest homing rates in males (63.1% ±2.4% s.e.m. inheritance), unaffected by parental effects, though the absence of gene drive activity in females (51.8% ±1.7% s.e.m. inheritance) may imply that there was simply insufficient nuclease expression in the female germline to produce a maternal effect (Fig 2H).

## New gene drive constructs confer significantly less fecundity costs in carrier females than previous constructs

Fecundity assays were performed to quantify the number of viable progeny (measured as larval output) in individual crosses of drive heterozygotes mated to wild type (Fig 2). Our previously published data for *vas2-CRISPR*<sup>h</sup> females showed vastly impaired fecundity (8% fecund, relative to wild type) in females heterozygous for this construct at the *AGAP007280* locus, despite this gene showing full haplosufficiency for this phenotype (i.e. individuals heterozygous for a simple null allele show normal fecundity) [7]. The most parsimonious explanation for this fitness effect is the (partial) conversion of the soma to the null phenotype, due to two sources: somatic 'leakiness' of the putative germline promoter driving the Cas9 nuclease and/or the stochastic distribution (mosaicism) across the soma of nuclease parentally deposited into the embryo.

Both the *zpg-CRISPR*<sup>h</sup> and the *nos-CRISPR*<sup>h</sup> drives showed a general marked improvement in relative female fecundity over the previous *vas2*-driven construct, with maximal fecundities (relative to wild type) of 54.9% and 55.5%, respectively (Fig 2E and 2G). Although we observed significant maternal deposition from the *nos-CRISPR*<sup>h</sup> construct, the reduction in fecundity of females that received the gene drive maternally was non-significant, compared to those that received the drive paternally (39.7% ±8.5% s.e.m. vs 55.5% ±7.3% s.e.m. larval output, respectively) (Fig 2G). For this construct we also noted a general reduction in male fecundity, regardless of paternal source of the gene drive allele. For *zpg-CRISPR*<sup>h</sup> we observed no reduction in male fecundity and the reduction in female fecundity was not subject to any maternal effect, consistent with the absence of maternal effect on homing for the same construct. Thus, the most likely explanation for the improved performance of the *zpg-CRISPR*<sup>h</sup> gene drive is the lack of a maternal effect, though the incomplete restoration of full fecundity in heterozygote females suggests there may still be some somatic leakiness.

The reason for the observed lower fecundity in *nos-CRISPR*<sup>h</sup> males is unexplained, given that the target gene affects female fertility. Possible explanations might include some unknown feature of the exogenous *nanos* promoter that affects sperm quality or general reduced fitness

resulting from the transgenesis procedure itself. We also note that the measure of fecundity in these experiments is marked by high inter-experimental variability and that, despite the inclusion of a wild-type control every time, we cannot exclude the possibility that batch effects can contribute in part to some of the differences observed between lines.

## Embryonic and germline rates of end-joining induced by gene drives containing *zpg* and *nos* promoters

Given its importance in the generation of resistant alleles, we designed a genetic screen to quantify the magnitude of parental nuclease deposition from each construct (Fig 3A). In this screen the wild-type target allele in the embryo, balanced against a pre-existing r1 allele (a 6bp GAGGAG deletion) is only exposed to a paternal or maternal dose of the nuclease in the absence of a genetically encoded drive construct. We performed amplicon sequencing of the relevant region of the *AGAP007280* target locus to sample and quantify the diversity of alleles at the target site. Two replicates were performed for each cross. In the absence of any Cas9 activity an equal ratio of the wild-type target site and the original r1 allele is expected among

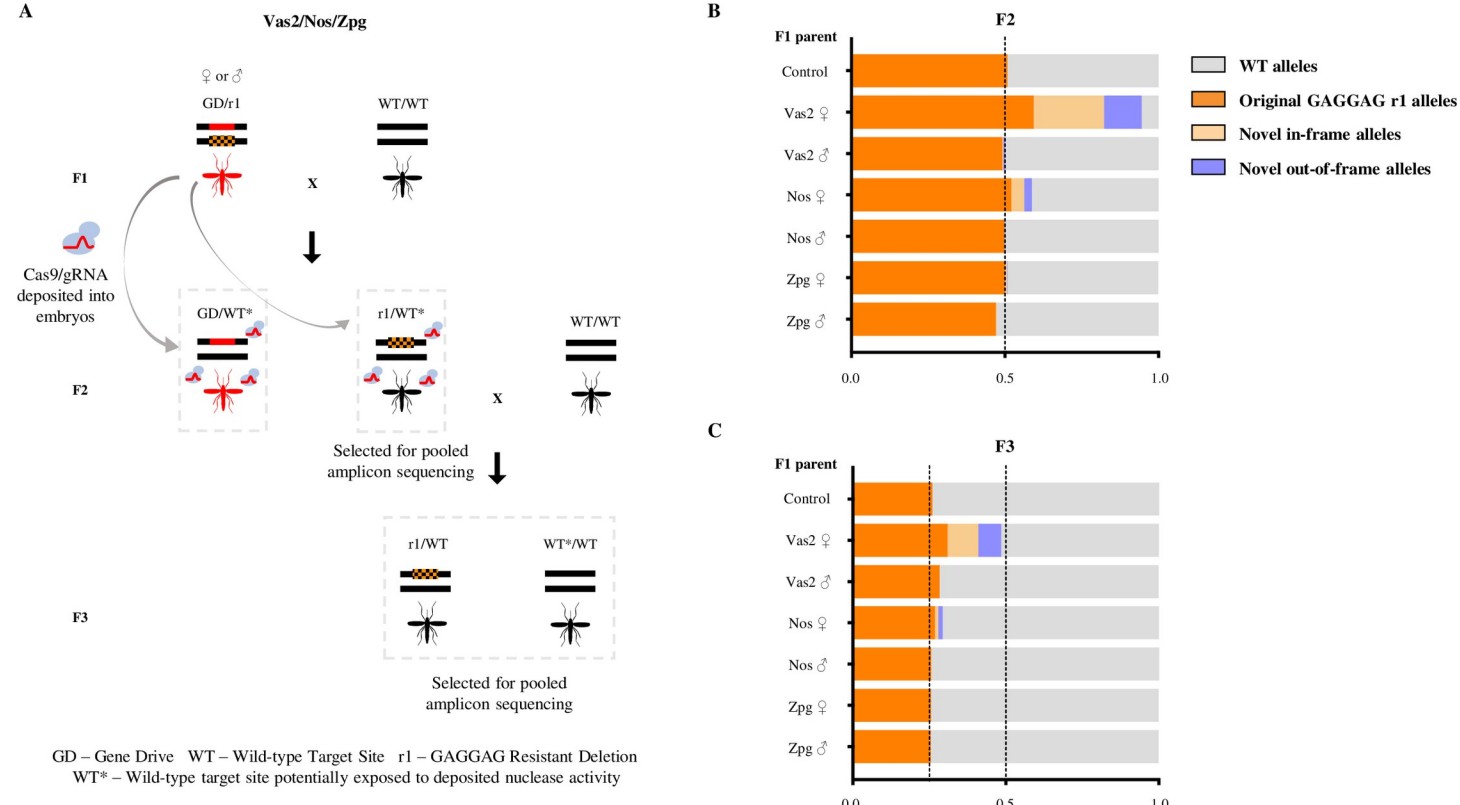

**Fig 3. Heritability and rates of formation of mutations caused by parentally deposited nuclease from different gene drive constructs.** (A) *zpg-CRISPR^h^*, *nos-CRISPR^h^* or *vas2-CRISPR^h^* were crossed to a strain homozygous for a defined r1 resistance allele (203-GAGGAG) to generate F1 heterozygotes of genotype GD/r1, containing both the gene drive allele (red bar) and the resistant allele (orange hatched bar). F1 heterozygotes and r1/r1 homozygotes (control) were crossed to wild type and their non-drive F2 progeny analysed by pooled amplicon sequencing across the target site in *AGAP007280*. The non-drive F2 progeny (r1*/wt) were crossed to wild type, and the entire F3 progeny was analysed by amplicon sequencing to determine the heritability of mutations caused by paternally-derived nuclease. (B) Amplicon sequencing results from F2 individuals (n>128) with genotype WT/r1 (selected by absence of dsRED-linked gene drive allele), whose only source of Cas9 nuclease would be from parental deposition into the embryo. In the absence of any deposited source of nuclease only the original r1 allele and the WT allele are expected, at a ratio of 1:1. Significant deviations from this ratio, or the presence of new target site mutations, therefore indicate the mutagenic activity of deposited Cas9. (C) Amplicon sequencing of F3 progeny deriving from those F2 crossed to wild type, in order to determine the heritability of mutations formed by parentally deposited nuclease. Mendelian inheritance of mutations present in the sampled F2 individuals would be expected to lead to a 2-fold reduction in their frequency between the F2 and F3 samples.

the chromosomes amplified. Novel indels arising at the target sequence are indicative of nuclease activity in the zygote as a result of parental deposition. The original r1 allele may also be over-represented among the F2 in this assay due to simple stochastic variation in its Mendelian inheritance or due to deposited Cas9 activity, either as a result of *de novo* generation through end-joining repair, or as a result of gene conversion through homing of the inherited r1 allele. Taking only the most conservative signal of embryonic end-joining (indels unique from the inherited r1 allele sequence), *vas2-CRISPR^h* generated high levels of maternally deposited Cas9 activity–affecting 70% of all nuclease-sensitive alleles (Figs 3B and S3). In the offspring of *nos-CRISPR^h* females the end-joining rates from maternal deposition were substantially lower (10.5% of nuclease-sensitive alleles), with no significant paternal deposition (Ordinary one-way ANOVA with Dunnett's multiple comparisons test, p = 0.9997). For the *zpg-CRISPR^h* line, we found no significant signal of maternal or paternal deposition (Ordinary one-way ANOVA with Dunnett's multiple comparisons test, p = 0.9564 for *zpg* females and p = 0.9996 for *zpg* males) (S3 Fig).

Mutant alleles generated in the embryo, if included in the germline tissue, will be resistant to subsequent homing during the production of gametes in the adult, reducing rates of drive transmission. Indeed, the gradation in terms of maternal contribution to end joining in the embryo (*vas2-CRISPR^h* > *nos-CRISPR^h* > *zpg-CRISPR^h*) mirrors the magnitude of the maternal effect observed in reducing drive transmission for each construct (Fig 2). Consistent with this, the presence in the F3 of novel end-joining mutations that were created in the F2 confirms that deposited nuclease affected also the germline tissue (Fig 3C). However, the frequency of resistant alleles generated in the F2 was approximately halved in the F3, consistent with their simple Mendelian inheritance. This is perhaps best seen by observing the frequency of the original r1 allele in the F2 and subsequently in the F3 of the *vas2-CRISPR^h* line. This indicates that any contribution of deposited nuclease to homing of these alleles in the germline is at best minimal, at least in our experimental setup. This is in contrast to reports in *Drosophila*, albeit in a split drive system with the gRNA cassette and the source of Cas9 on separate constructs, where use of the *nanos* promoter can lead to 'shadow drive' in which perduring maternally deposited Cas9 can cause homing of the gRNA cassette in the germline even in the absence of genetically encoded Cas9 [30,31].

Given the improved characteristics of the *zpg-CRISPR^h* construct in terms of fecundity and reduced embryonic end-joining we also investigated its propensity to generate end-joining mutants in the germline, a feature that has been observed previously both in mosquitoes and *Drosophila* [9,13,32]. We designed a genetic cross that allowed us to enrich and capture those chromosomes (~2–3%) that were subjected to the nuclease activity of the *zpg-CRISPR^h* gene drive in the adult germline but were not 'homed' (Fig 4). Interestingly from males (n = 239) we noticed a much greater heterogeneity of alleles (82 mutated chromosomes covered by 18 distinct alleles) among those generated from end-joining repair than we did in females (n = 218), where the vast majority (88%) of non-homed chromosomes were mutated but these comprised just 4 unique alleles. The apparent differences in the behaviour of this gene drive between the male and female germlines may reflect differences in the timing of nuclease activity and chromosomal repair in the two sexes. It may also suggest that early repair events in the female germline, when only a few germline stem cells are present and indeed only one of 16 germline cells becomes an oocyte, leads to a clustering of repair events among the eventual gametes [10,13,32].

## *zpg-CRISPR^h* spreads close to fixation in caged releases and exerts a large reproductive load on the population

Given its improved fecundity and its lower mutagenic activity we investigated the potential for the *zpg-CRISPR^h* gene drive to spread throughout naïve mosquito populations. Two replicate

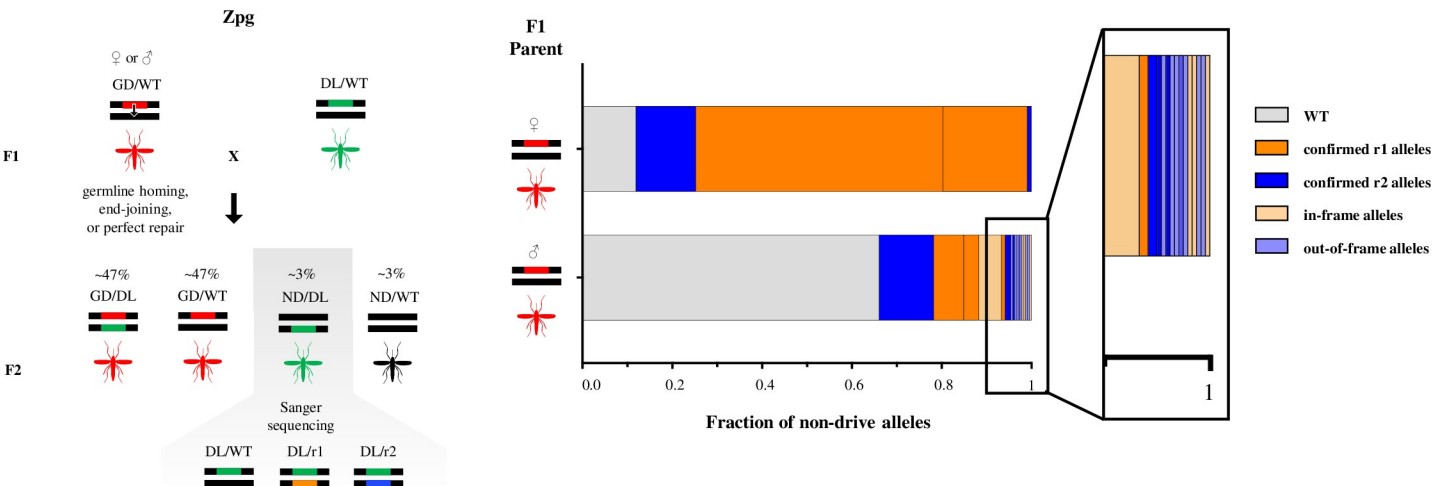

**Fig 4. The rate and outcome of meiotic end-joining (EJ) differs in the male and female germline.** F1 females (n>500) or males (n>200) heterozygous for the gene drive allele (linked to RFP) (GD) were crossed to a GFP-linked 'docking line' (DL). The docking line contains a null allele (DL) of the target gene, generated by inserting a full GFP marker cassette at the target site, thereby both truncating the *AGAP007280* coding sequence and ablating the target site recognised by the gene drive, while allowing easy visual tracking of this allele. The rare fraction (~3%) of F2 progeny that had inherited a DL but not a GD allele were selected and their non-drive allele (ND), that had been meiotically exposed to nuclease activity, was sequenced to reveal the frequency and nature of EJ mutations taking place during meiosis (offspring from females n = 218; offspring from males n = 239). The frequency of all WT unmodified alleles is shown in grey. Mutant alleles, whose effect had previously been tested by balancing against a null allele, were classified as 'confirmed r1' (orange) or 'confirmed r2' (blue) and novel mutations, whose phenotype has not yet been definitively proved, as in-frame (light orange) or out-of-frame (light blue).

cages were initiated with either 10% or 50% starting frequency of drive heterozygotes, and monitored for 16 generations, which included pooled sequencing of the target locus at various generations. The drive spread rapidly in all four trials, to more than 97% of the population, achieving maximal frequency in just 4–10 generations (Fig 5A). In each trial, the drive sustained more than 95% frequency for at least 3 generations before its spread was reversed by the gradual selection of drive-resistant alleles. Notably, we observed similar dynamics of spread, in terms of rate of increase and duration of maximal frequency, whether the gene drive was released at 50% or 10%, demonstrating that initial release frequency has little impact (providing it is not stochastically lost initially) upon the potential to spread; i.e. a release of low numbers of mosquitoes with the gene drive can be as effective (if not as quick) at reaching high frequency.

The rates of invasion observed here represent a significant improvement over the first generation gene drive (*vas2-CRISPR^h*) targeted to the exact same target sequence (included for comparison in Fig 5A), where the spread of the drive was markedly slower and declined before it reached 80% frequency in the population [7].

Our population suppression gene drive is designed to exert a reproductive load on the population by targeting a gene that is essential for viable egg production. To investigate the magnitude of this load, we counted (from generation 4) the number of viable eggs produced each generation to measure the level of population suppression. Egg production was suppressed by an average of 92% (compared to maximal output of the cage) in each cage at or shortly after the peak in drive frequency, representing a reduction from more than 15,000 eggs to under 1,200 eggs (Fig 5A). Since we set a fixed carrying capacity for the adult population in the cage, selecting 600 larvae each generation, the reduction in egg output, although severe, was insufficient to suppress the population below this carrying capacity. In practice, in the wild, the effect

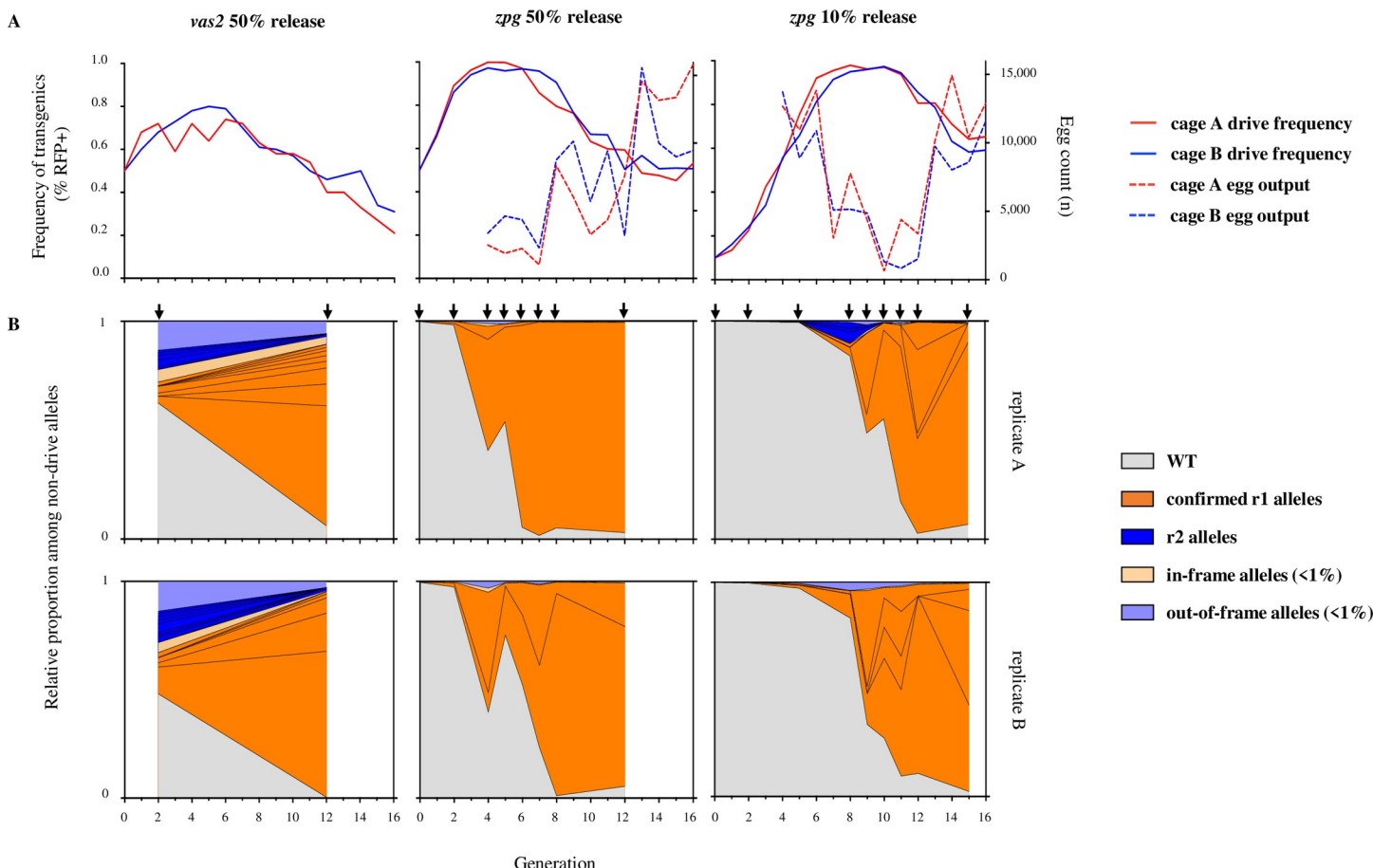

**Fig 5. *zpg-CRISPR^h* rates of spread through caged populations, reproductive load and real-time profiling of resistance.** *zpg-CRISPR^h*/+ were released into replicate caged WT populations at 10% (right) or 50% (middle) *CRISPR^h*/+ initial frequency. (A) The frequency of drive-modified mosquitoes (solid line) was recorded each generation by screening larval progeny for presence of DsRed linked to the *CRISPR^h* allele. Entire egg output per generation was also recorded (dashed line). *zpg-CRISPR^h* drive spread was compared to previous results for *vas2-CRISPR^h* (left panel) from Hammond *et al.* (2017). (B) The nature and frequency of non-drive alleles was determined for several generations (arrows) of the caged population experiments by amplicon sequencing across the target site at *AGAP007280* in pooled samples. Vertically aligned panels represent the two replicates from each treatment. The frequency of all WT alleles is shown in grey. Mutant alleles individually exceeding 1% frequency in any generation were labelled as 'confirmed r1 alleles' (orange) or 'r2 alleles' (blue), all had been previously confirmed as r1 or r2. Mutations that were individually present at less than 1% frequency were grouped together as in-frame (light orange) or out-of-frame (light blue).

of these levels of reproductive suppression will depend very much on the strength of density-dependent mortality that is occurring in the relevant mosquito larval habitats [33,34].

## The creation and selection of resistant mutations is delayed in cage invasion experiments when using the *zpg* promoter

Both r1 mutations and r2 mutations can retard the invasiveness of a gene drive, yet r1 mutations are more problematic since they are likely to be strongly selected due to the relative positive fitness they confer by restoring function to the target gene.

To determine the rate of accumulation of these mutations during the invasion experiment we sequenced across the target locus in samples of pooled individuals from intermittent generations that spanned the rise and fall in frequency of the gene drive. We recorded the relative proportion of wild-type alleles and various r1 and r2 alleles among remaining non-drive alleles as the gene drive invaded the population over time (Fig 5B). For the *zpg-CRISPR^h* constructs, which showed the biggest improvement in fecundity of carrier females (containing one drive

allele and one wild-type allele) compared to previous constructs, resistance did not show obvious signs of selection (judged by large increases in frequency) until more than 90% of individuals were positive for the gene drive. In contrast, our previous data for the *vas2*-driven constructs showed that resistance was selected much earlier, being already present at high frequency in the 2nd generation when the frequency of drive-positive individuals was less than 70%. This is likely due to a combination of factors: the somatic leakiness of the *vas2* promoter resulting in a significant fitness cost in carrier females and thus a relative fitness benefit to the r1 allele at a much earlier stage in the invasion, even when drive homozygotes are still rare; nuclease deposition from the *vas2* promoter leading to a high initial frequency of r2 alleles, resulting in many individuals that are null (either homozygous for the r2 allele or contain one r2 allele and one drive allele) for the target gene therefore meaning that any non-null allele has a stronger relative fitness advantage.

Not only does the *zpg* promoter delay the onset of drive resistance, it also reduces the range of resistant alleles created (Fig 5B). In the 50% release cages, only 2 mutant alleles were detected above threshold frequency (>1% of non-drive alleles) both of which had been previously confirmed to be of the r1 class. By generation 8, one of the two had spread to more than 90% frequency yet the most abundant allele was reversed in the replicate cages–suggesting that selection for one or the other resistant mutations is stochastic and not because one is more effective at restoring fertility. The equivalent release of *vas2-CRISPR*[h] generated 9–12 mutant alleles above 1% frequency by generation 2 and this variance was maintained over time despite a strong stratification towards those of the r1 class [9]. This reduced complexity of alleles generated by *zpg-CRISPR*[h] may be related to the late onset of resistance, when the majority of alleles at the target locus are drive alleles meaning that there are very few 'free' alleles on which r1 alleles can be generated, resulting in the stochastic selection of just a few.

The very late onset of resistance–if it is to occur–is a feature, not immediately intuitive, of this type of gene drive whose goal is suppression and where heterozygous 'carriers' are intended to show minimal fitness costs. Selection for r1 alleles only really becomes strong when the gene drive is close to fixation; when most individuals have at least one copy of the gene drive the relative advantage of an r1 allele over remaining wild-type alleles (that have a high probability of being removed by homing of the gene drive allele) or the drive allele (with high probability of finding itself in a barren female, homozygous for the drive) is at its highest.

## Incorporating experimental rates of resistance, embryonic end-joining and homology-directed repair into a population model

We extend previous models of homing-based gene drives [9,35,36], allowing the option of embryonic activity from paternally and maternally derived nuclease that can be resolved through end-joining–forming r1 and r2 alleles–or HDR. Using this model (see S1 Text) and the baseline parameter values from rates of fecundity and repair outcomes following germline and embryonic nuclease activity (S3 and S4 Tables) we generated the expected dynamics of spread and population egg output over the 16 generations of the experiment (Fig 6). For both 50% and 10% releases, the model captures the general trend of initial spread that is met by a significant suppression in egg output and an eventual recovery of the population as the gene drive is replaced by resistant mutations. Notably, the observed dynamics of spread and population suppression were faster than the deterministic prediction in all replicates. One possible explanation for this is the sensitivity of the model to the fertility of female *zpg-CRISPR*[h] heterozygotes and the extent to which fertility is restored by resistant R1 alleles (S2 Fig)–two parameters that are particularly difficult to measure precisely from our simple measurements of female fecundity and whose real values may be somewhat different in the inter-generational

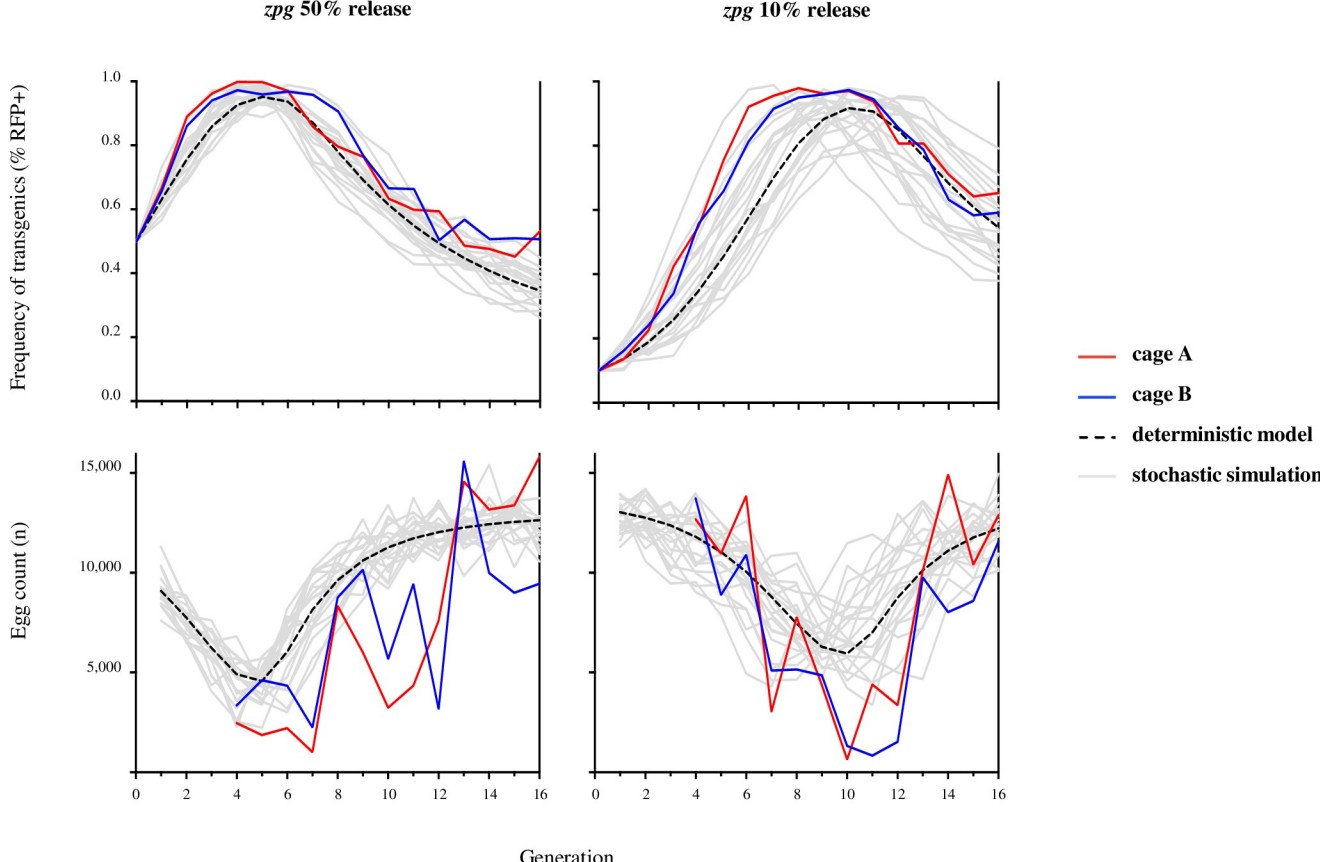

**Fig 6. Comparison of observed *zpg-CRISPR^h* drive frequency and egg output data to deterministic and stochastic model predictions.** *zpg-CRISPR^h/+* individuals were released into replicate caged WT populations at 50% (left) or 10% (right) initial frequency of *CRISPR^h/+*. The frequency of drive-modified mosquitoes (top panels) and the entire egg output per generation (bottom panels) were recorded. Observed data were compared to a deterministic, discrete-generation model (dashed line) based upon observed rates of fertility, homing, and resistance generated by end-joining in the germline and embryo (S1 Table). Grey lines show 20 stochastic simulations assuming that females may fail to mate, or mate once with a male of a given genotype according to its frequency in the male population; mated females produce eggs in amounts determined by sampling with replacement from experimental values that subsequently hatch according to the genotype of the mother; of the 600 larvae chosen randomly to populate next generation, some fail to survive to adulthood. Probabilities of mating, egg production, hatching and survival from larval stage to adult are estimated from experiment (S2 Table) and random numbers for these events drawn from the appropriate multinomial distributions. Model egg counts for each release are normalised such that the deterministic value corresponds to the average of the two replicate cage experiments at generation sixteen.

cage invasion experiment [7,9]. Interestingly, these parameters exert their effect upon drive dynamics in concert: improvements to heterozygous female fertility increase the rate of spread and peak in drive frequency, whereas an r1 allele that fails to restore full wild-type fertility retards the rate at which the drive is removed from the population. Alternatively, there may be extra sources of stochastic variation such as variation in distributions of egg number and degree of participation of females and males in mating not captured in the model.

## Conclusions

Building suppression gene drives for mosquito control that are fit for purpose requires a combination of the following features: optimal target sites that show high functional constraint; the targeting of multiple sites by the same gene drive construct; fine tuning of the expression of the nuclease that serves as the gene drive's 'motor' in order that the drive shows the most efficient invasion dynamics. A number of these theorised improvements to design have been

substantiated: multiplexing through the use of multiple guide RNAs in a single drive construct has been shown to improve robustness of a range of drive elements [13,21,37,38]; judicious choice of functionally constrained sites in essential genes has meant that gene drives targeting such sites do not generate resistant alleles that are selected, at least in the laboratory [8,37]. For the last aspect–optimising expression–we have shown here that simple changes to the promoter controlling nuclease activity can lead to drastic improvements in the speed of invasion due to two principal reasons: 1) less propensity to induce end-joining repair in the germline and 2) improved fecundity of 'carrier' females that contain a single copy of the gene drive and transmit it in a super-Mendelian fashion to their offspring.

The combined effect of the improvements conferred by the *zpg* promoter meant that a gene drive targeting a female fertility gene, at a site otherwise extremely prone to generate resistance [9], could still spread close to fixation while imposing a very strong reproductive load before resistant alleles eventually became selected.

The propensity for maternal and paternal deposition of the nuclease under control of the promoters of the drive constructs is an important variable that we have shown can vary greatly. There is also the possibility that both the end-joining/HDR ratio and the magnitude of parental effect might additionally be locus dependent to some extent [13,30]. Indeed, when the *zpg* promoter was used to control expression of a homing-based gene drive at the *doublesex* locus we noticed a paternal effect contributing to reduced fecundity that was not observed here [8]. Thus, we cannot exclude the possibility that local cross-talk with other nearby transcriptional enhancers or promoters, either endogenous to the genome or as part of additional elements of any gene drive construct, may affect its expression characteristics.

Our data on the relative contributions of end-joining repair and homology-directed repair in the germline and in the early embryo will be particularly important for the modelling of expected performance of gene drive constructs in mosquito populations, not only for suppressive drives but for so-called population replacement drives that also rely on CRISPR-based homing activity [6,39]. This data, coupled with the improvements to drive performance conferred by promoter choice, should help expedite the building of effective gene drives for mosquito control.

## Methods

### Ethics statement

All animal work was conducted according to UK Home Office Regulations and approved under Home Office License PPL 70/6453.

### Amplification of promoter and terminator sequences

In determining the length of upstream region to take from each candidate gene to serve as a promoter we took a maximal region of 2kb upstream of the coding sequence, or the entire intergenic distance to the end of the coding sequence of the neighbouring gene, if shorter. The published *Anopheles gambiae* genome sequence provided in Vectorbase [39] was used as a reference to design primers in order to amplify the promoters and terminators of the three *Anopheles gambiae* genes: *AGAP006098* (*nanos*), *AGAP006241* (*zero population growth*) and *AGAP007365* (*exuperantia*). Using the primers described in S5 Table we performed PCRs on 40 ng of genomic material extracted from wild-type mosquitoes of the G3 strain using the Wizard Genomic DNA purification kit (Promega). The primers were modified to contain suitable Gibson assembly overhangs (underlined) for subsequent vector assembly, and an XhoI restriction site upstream of the start codon. Promoter and terminator fragments were 2092 bp and

601 bp for *nos*, 1074 bp and 1034 bp for *zpg*, and 849 bp and 1173 bp for *exu*, respectively. The sequences of all regulatory fragments can be found in S6 Table.

### Generation of CRISPR[h] drive constructs

We modified available template plasmids used previously in Hammond *et al.* (2016) to replace and test alternative promoters and terminators for expressing the Cas9 protein in the germline of the mosquito. p16501, which was used in that study carried a human-optimised Cas9 (hCas9) under the control of the *vas2* promoter and terminator, an RFP cassette under the control of the neuronal *3xP3* promoter and a U6:sgRNA cassette targeting the *AGAP007280* gene in *Anopheles gambiae*.

The hCas9 fragment and backbone (sequence containing 3xP3::RFP and a U6::gRNA cassette), were excised from plasmid p16501 using the restriction enzymes XhoI+PacI and AscI+AgeI respectively. Gel electrophoresis fragments were then re-assembled with PCR amplified promoter and terminator sequences of *zpg*, *nos* or *exu* by Gibson assembly to create new CRISPR[h] vectors named p17301 *(nos)*, p17401 *(zpg)* and p17501 *(exu)*.

### Integration of gene drive constructs at the *AGAP007280* locus

CRISPR[h] constructs containing Cas9 under control of the *zpg*, *nos* and *exu* promoters were inserted into an *hdrGFP* docking site previously generated at the target site in *AGAP007280* (Hammond *et al.*, 2016). Briefly, *Anopheles gambiae* mosquitoes of the *hdrGFP-7280* strain were reared under standard conditions of 80% relative humidity and 28°C, and freshly laid embryos used for microinjections as described before [40]. Recombinase-mediated cassette exchange (RMCE) reactions were performed by injecting each of the new CRISPR[h] constructs into embryos of the *hdrGFP* docking line that was previously generated at the target site in *AGAP007280* [7]. For each construct, embryos were injected with solution containing CRISPR[h] (400ng/µl) and a *vas2*::*integrase* helper plasmid (400ng/µl) [41]. Surviving $G_0$ larvae were crossed to wild type and transformants were identified by a change from GFP (present in the *hdrGFP* docking site) to DsRed linked to the CRISPR[h] construct that should indicate successful RMCE.

### Phenotypic assays to measure fertility and rates of homing

Heterozygous CRISPR[h]/+ mosquitoes from each of the three new lines *zpg-CRISPR[h]*, *nos-CRISPR[h]*, *exu-CRISPR[h]*, were mated to an equal number of wild-type mosquitoes for 5 days in reciprocal male and female crosses. Females were blood fed on anesthetized mice on the sixth day and after 3 days, a minimum of 40 were allowed to lay individually into a 25-ml cup filled with water and lined with filter paper. The entire larval progeny of each individual was counted and a minimum of 50 larvae were screened to determine the frequency of the DsRed that is linked to the CRISPR[h] allele by using a Nikon inverted fluorescence microscope (Eclipse TE200). When testing each line, age matched wild-type mosquitoes were included as comparators to control for variability in bloodmeal batch/insectary conditions that can affect clutch size. Females that failed to give progeny and had no evidence of sperm in their spermathecae were excluded from the analysis. Statistical differences between genotypes were assessed using the Kruskal-Wallis and Mann-Whitney tests, performed using GraphPad Prism version 8.4.3 (471) for Mac, GraphPad Software, San Diego, California USA, www.graphpad.com. The entirety of the raw data including full list of test statistics and sample sizes, as well all raw experimental data points, are included in S1 Data as an Excel file with the data tabulated according to the experimental figure in which it was included.

## Caged population invasion experiments

The cage trials were performed following the same principle described before in Hammond *et al.* (2016). Briefly, heterozygous *zpg-CRISPR*[h] that had inherited the drive from a female parent were mixed with age-matched wild type at L1 at 10% or 50% frequency of heterozygotes. At the pupal stage, 600 were selected to initiate replicate cages for each initial release frequency. Adult mosquitoes were left to mate for 5 days before they were blood fed on anesthetized mice. Two days after, the mosquitoes were left to lay in a 300ml egg bowl filled with water and lined with filter paper. Each generation, all eggs were allowed two days to hatch and 600 randomly selected larvae were screened to determine the transgenic rate by presence of DsRed and then used to seed the next generation. From generation 4 onwards, adults were blood-fed a second time and the entire egg output photographed and counted using JMicroVision V1.27. Larvae were reared in 2L trays in 500 ml of water, allowing a density of 200 larvae per tray. After recovering progeny, the entire adult population was collected and entire samples from generations 0, 2, 4, 5, 6, 7, 8 and 12 (50% release) and 0, 2, 5, 8, 9, 10, 11, 12 and 15 (10% release) were used for pooled amplicon sequence analysis.

## Pooled amplicon sequencing

Pooled amplicon sequencing was performed essentially as described before in Hammond *et al.* (2017). Genomic DNA was mass extracted from pooled samples of mosquitoes using the Wizard Genomic DNA purification kit (Promega), and 90 ng of each used for PCR using KAPA HiFi HotStart Ready Mix PCR kit (Kapa Biosystems) in 50 ul reactions. For caged experiment generations 0, 2, 5 & 8, a 332-bp locus spanning the target site was amplified using primers designed to include the Illumina Nextera Transposase Adapters (underlined), 7280-Illumina-F (TCGTCGGCAGCGTCAGATGTGTATAAGAGACAGGGAGAAGGTAAA TGCGCCAC) and 7280-Illumina-R (GTCTCGTGGGCTCGGAGATGTGTATAAGAGA CAGGCGCTTCTACACTCGCTTCT). For caged experiment generations 4, 6, 7 and 12 of the 50% release; and 9, 10, 11, 12 and 15 of the 10% release, a 196-bp locus spanning the target site was amplified using primers designed to include Illumina partial adapters (underlined), Illumina-AmpEZ-7280-F1 (ACACTCTTTCCCTACACGACGCTCTTCCGATCTCGTTAAC TGTCTTGGTGGTGAGG) and Illumina-AmpEZ-7280-R1 (GACTGGAGTTCAGACGTG TGCTCTTCCGATCTCACGCTTAACGTCGTCGTTTC). For deposition testing, a 200-bp locus spanning the target site was amplified using primers designed to include Illumina partial adapters (underlined), Illumina-AmpEZ-7280-F2 (ACACTCTTTCCCTACACGACGCTCT TCCGATCTCGGGCAAGAAGTGTAACGG) and Illumina-AmpEZ-7280-R2 (TGGAGTT CAGACGTGTGCTCTTCCGATCTGTCGTTTCTTCCGATGTGAAC). PCR reactions were amplified for 20 cycles and subsequently processed and sequenced using an Illumina MiSeq instrument (Genewiz).

## Analysis of pooled amplicon sequencing

Pooled amplicon sequencing reads were analysed using CRISPResso software v1.0.8 [42] using an average read quality threshold of 30. Insertions and deletions were included if they altered a window of 20 bp surrounding the cleavage site that was chosen on the basis of previously observed mutations at this locus [9]. Allele frequencies were calculated by summing individual insertion or deletion events across all haplotypes on which they were found. A large insertion event, representing incomplete homing of *CRISPR*[h], was found to occur outside of this window and its combined frequency across several haplotypes was calculated and included in the final frequency tables.

## Deposition testing

F1 heterozygotes containing a gene drive and resistant allele (GD/r1) were generated by crossing *zpg-CRISPR^h*, *nos-CRISPR^h* or *vas2-CRISPR^h* to a resistant strain that is homozygous for the 203-GAGGAG r1 allele at the target site in *AGAP007280*. This scheme allowed the testing of all gene drive constructs, including those (e.g. *vas2-CRISPR^h*) that would otherwise cause high levels of female sterility when balanced against a wild-type allele. An average of 40 F1 heterozygotes were group mated to an excess of wild type in reciprocal male and female crosses, and allowed to lay *en masse*. F2 progeny were screened for the absence of DsRed that is linked to the *CRISPR^h* allele and pooled together for mass genomic DNA extraction and pooled amplicon sequencing as described elsewhere. A minimum of 118 F2 individuals were interrogated for each condition. The summary data for amplicon sequencing experiments reported for Fig 3 (testing of deposited nuclease activity) is available in S2 Data.

## Estimation of meiotic end-joining

A minimum of 200 male or 500 female *zpg-CRISPR^h* drive heterozygotes (F0) were crossed to a line ('docking line') heterozygous for a GFP⁺ marked knock-out *AGAP007280* allele and allowed to lay eggs. Upon egg-hatching, L1 larvae were sorted using COPAS (complex object parametric analyser and sorter), as in Marois *et al.* [43]. The very rare fraction of progeny (F1) that inherited the GFP⁺ marked knockout but not the *zpg-CRISPR^h* allele due to lack of homing, were isolated from the progeny pool. They were grown to adulthood and their gDNA was individually extracted using either the Wizard Genomic DNA purification kit (Promega) or the DNEasy Blood & Tissue Kit (Qiagen). A 1048-bp region spanning the gene drive target site was amplified using primers Seq-7280-F (GCACAAATCCGATCGTGACA) and Seq-7280-R3 (GGCTTCCAGTGGCAGTTCCGTA) and Sanger-sequenced using primer Seq-7280-F5 (CGTTTGTGTGTCAGAGCAAGTCG), so that only the allele that was exposed to prior nuclease activity meiotically would amplify (and not the GFP⁺ allele). In total, 218 F1 progeny descended from female *zpg-CRISPR^h* heterozygote and 239 F1 progeny descended from male *zpg-CRISPR^h* heterozygote were analysed.

## Supporting information

**S1 Fig. Cleavage by *CRISPR^h* can generate resistant mutations as a by-product of error-prone end-joining.** After cleavage by the nuclease, the majority of target sites will be repaired by homology-directed repair (HDR), leading to a copying over of the *CRISPR^h* allele called homing. A small fraction of targets may remain unmodified or may repair perfectly, resulting in a target that can be cleaved upon re-exposure by the nuclease. Chromosomes that are repaired by end-joining may generate a mutant target site that can no longer be cleaved by the nuclease. If the target site is essential (i.e. a female fertility gene), then a mutation that disrupts the function of the target gene, called an r2 mutation, will be selected out of the population. Mutations that re-code a functional target gene, called an r1 mutation, are resistant to the gene drive and will come under strong selection in presence of the drive.
(TIF)

**S2 Fig. Comparison of cage data with deterministic model results incorporating percentage change in parameter estimates.** (A) Increasing the fertility of heterozygous *zpg-CRISPR^h* females by up to 80% can accurately predict the experimental data, suggesting the fertility assays underestimate their true fecundity. Note that for (A), W/D female fertilities with maternal/paternal parental effects are varied together, keeping their ratio constant. (B) A small increase in the homing rates of both males and females can have a dramatic effect on the

spread of the gene drive during the early generations. By increasing the homing rate estimates of males, similar to that of females, the improved rate of spread better resembles experimental observation. (C) The resilience of the gene drive within the populations increases when the fertility of females with at least one resistant functional allele (r1) is reduced. The reduced output in progeny of these females slows the spread of r1 alleles and thus, allows the drive to spread closer to fixation and to exert a longer lasting and stronger reproductive load on the population. These modelling data suggest that small cage experiments and phenotypic assays provide crude estimates of fertility and drive, and thus both modelling and large-cage testing are needed to better capture and estimate strain performance.
(TIF)

**S3 Fig. Depletion of wild-type target site alleles and formation of mutations caused by parentally deposited nuclease from different gene drive constructs.** *zpg-CRISPR$^h$* ('Zpg'), *nos-CRISPR$^h$* ('Nos')or *vas2-CRISPR$^h$* ('Vas') were crossed to a resistant strain (homozygous for the r1 allele 203-GAGGAG) to generate F1 heterozygotes containing both a gene drive and resistant allele (GD/r1). F1 heterozygote males (M) and females (F) and r1/r1 homozygote females (control) were crossed to wild type and their non-drive F2 progeny analysed by pooled amplicon sequencing across the target site in *AGAP007280*. (A) Amplicon sequencing results from F2 individuals with genotype WT/r1 (left) and their F3 progeny derived from the F2 crossed to WT (right) are plotted to show the fraction of novel resistant ("R") alleles ('Proportion of novel R', top). (B) Amplicon sequencing results from F2 individuals with genotype WT/r1 (left) and their F3 progeny derived from the F2 crossed to WT (right) are plotted to show the fraction of WT alleles ('Depletion of WT ', middle). Frequencies expected by Mendelian inheritance of the WT allele are indicated (dashed line). Deposited nuclease significantly depleted WT and generated novel R for *nos-CRISPR$^h$* and *vas2-CRISPR$^h$* females in the F2, but not *zpg-CRISPR$^h$* females, or males of any class (Ordinary one-way ANOVA with Dunnett's multiple comparisons test, ****: $p < 0.0001$, ***: $p < 0.001$, *: $p < 0.05$, ns: non-significant). (C) Rates of embryonic EJ were estimated based upon the relative frequencies of novel R alleles in the F2 (left) and F3 (right). F2 estimates were calculated from the frequency of novel R alleles amongst the sum of WT and novel R (left). F3 estimates were calculated by multiplying the frequency of novel R by 4, the dilution factor expected due to additional WT alleles received from the parents (right). A minimum of 128 individuals were sequenced for each condition in the F2 and F3 generations and the number of founder F1 individuals is indicated (n). A summary of this data is shown in Fig 3B and 3C.
(TIF)

**S1 Data. Raw Data for Figs 2–6 Excel file, tabulated according to figure.** Includes all raw data points, lists of test statistics and sample sizes.
(XLSX)

**S2 Data. Summary data for amplicon sequencing experiments reported for Fig 3 (testing of deposited nuclease activity).**
(XLSX)

**S1 Text. Supplementary Methods describing the mathematical modelling in detail.**
(DOCX)

**S1 Table. Model parameters for *zpg-CRISPR$^h$*.**
(DOCX)

**S2 Table. Additional parameters for stochastic model for *zpg-CRISPR$^h$*.**
(DOCX)

**S3 Table. Primers used in this study to assemble the transformation vectors.**
(DOCX)

**S4 Table. Promoter and terminator sequences used in gene drive constructs.**
(DOCX)

**S5 Table. The proportions of germline stem cells of different genotypes arising from paternal effects.**
(DOCX)

**S6 Table. Genotype fitnesses and the proportion of each type of gamete produced by each genotype.**
(DOCX)

## Author Contributions

**Conceptualization:** Andrew Hammond, Tony Nolan.

**Data curation:** Andrew Hammond, Xenia Karlsson, Ioanna Morianou, Andrea Beaghton, Nace Kranjc, Tony Nolan.

**Formal analysis:** Andrew Hammond, Andrea Beaghton, Nace Kranjc, Tony Nolan.

**Funding acquisition:** Austin Burt, Andrea Crisanti, Tony Nolan.

**Investigation:** Andrew Hammond, Xenia Karlsson, Ioanna Morianou, Roberto Galizi, Tony Nolan.

**Methodology:** Andrew Hammond, Xenia Karlsson, Ioanna Morianou, Kyros Kyrou, Tony Nolan.

**Resources:** Kyros Kyrou, Matthew Gribble, Andrea Crisanti, Tony Nolan.

**Supervision:** Austin Burt, Andrea Crisanti, Tony Nolan.

**Visualization:** Tony Nolan.

**Writing – original draft:** Andrew Hammond, Tony Nolan.

**Writing – review & editing:** Andrew Hammond, Ioanna Morianou, Tony Nolan.

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
