## [Decision Letter · Decision Letter 0]

2 Oct 2020

Dear Dr Nolan,

Thank you very much for submitting your Research Article entitled 'Regulating the expression of gene drives is key to increasing their invasive potential and the mitigation of resistance' to PLOS Genetics. Your manuscript was fully evaluated at the editorial level and by independent peer reviewers. The reviewers appreciated the attention to an important topic but identified some aspects of the manuscript that should be improved.

We therefore ask you to modify the manuscript according to the review recommendations before we can consider your manuscript for acceptance. Your revisions should address the specific points made by each reviewer.

[LINK]

Yours sincerely,

Owain Rhys Edwards, Ph.D.

Guest Editor

PLOS Genetics

Gregory Barsh

Editor-in-Chief

PLOS Genetics

Reviewer's Responses to Questions

**Comments to the Authors:**

Reviewer #1: Gene drive is being used for population modification (replacement/alteration) and population suppression. Much recent success revolves around the use of Cas9 to site-specifically cleave one or more target sites, with the outcome of repair following cleavage then determining, depending on the system, whether drive occurs or not. In all Cas9-based drive systems (other than underdominance) germline cleavage is required. Most of the promoters used to drive expression in the germline have effects on other life stages, either through maternal carryover-dependent cleavage of the paternal chromosome in somatic cells, and/or through cleavage in somatic cells due to a non-germline component of gene expression. For those of us working on purely cleavage-based mechanisms of Cas9-mediated drive such as clvr, expression at other stages is generally not problematic. In fact maternal carryover-dependent cleavage makes drive stronger. However, in the context of homing, particularly population suppression, maternal carryover and expression in somatic cells can be devastating for several reasons. First, since the early embryo lacks G1 and G2 cycles (at least in Drosophila there is only S and M for the first 12 divisions), there is no possibility of repair by HR. This leads to a high frequency of repair by NHEJ, which can create resistant alleles. Second, cleavage in the zygote makes individuals heterozygous for the HEG unfit. The HEG insertion disrupts one allele of the essential gene targeted. In the absence of carryover or somatic expression these individuals are fit since the genes being targeted are haplosufficient. However, when carryover or somatic expression disrupts the second allele, heterozygotes now suffer a fitness cost due to being homozygous loss of function (LOF) in many cells. this can slow or stop drive.

This group has addressed many of these issues by targeting a female-splice form of the doublesex (dsx) gene, which apparently cannot (thus far) mutate to a functional but uncleavable (resistant) form. This solves the problem of carryover-dependent NHEJ. They have also identified a germline promoter, from the zpg gene, that has minimal (though not completely absent) maternal carryover, limiting the fitness costs to heterozygotes. Work describing the use of this promoter (and the associated HEG) to bring about complete population elimination has been described in two recent papers, Kyrou et al., 2018, and Simoni et al., 2020.

What has not been reported is a careful description of how the authors characterized zpg-HEGs and those of earlier generations, which fared less well, exactly how non-optimal characteristics can lead to failure, and how (quantitatively) constructs that avoid these pitfalls do so. These details are critical as they inform the field about how drives can and will fail, and how to characterize them so as to understand where and when this will occur. Thus, while this manuscript comes after reports of dramatic success in bringing about suppression, it is a very important work in its own right. It provides a careful blueprint for understanding how a HEG should be characterized: how to "look under the hood" using genetics and sequencing to determine exactly what is going on with the parts that make up the HEG.

The strategy taken is interesting, and involves targeting a gene, nudel in Drosophila, required for female fertility. The authors had previously targeted this gene only to have resistant alleles arise rapidly (earlier publications). They target it again here precisely because it fails, and thus the rate of failure and how failure occurs can provide insight into how to avoid it (dsx, because it always works does not provide this opportunity). HEGs were constructed that target nudel, with Cas9 being driven by one of three different germline promoters, zpg, nos and exu. Data from the earlier published experiments using the vasa promoter served as a benchmark for improvements.

Overall the experiments are very clear and the data convincing. The use of marked chromosomes to follow the generation and fate of alleles over multiple generations is really a tour de force of population genetic characterization of an evolving system, particularly in a non-model system. The conclusions are clear and provide a clear roadmap for others in terms of how to characterize HEG behavior. This work will become and set the standard for all work going forward– which is a good thing.

I have a series of comments below, but all of these are questions or comments asking for further clarification of some discussion or figures.

1. One thing I am unclear on in Fig. 2CEGI is what the marks are in the top row of each plot. I am guessing these may be fecundity plots for wildtype females crossed to wildtype males, since then the statistical significance numbers make sense. This is suggested by the WT symbol on the far left, but, it is not indicated explicitly in the figure legend.

Also, throughout the figures, including Fig. 2, the type used is very thin and not bolded. it would help a lot if all type and male and female symbols were bolded and in some cases perhaps made larger. There is plenty of space in the figure. The same goes for the schematic presented in panel A. All parts could be larger and more clear. The authors might also consider expanding the schematic a bit to try and capture some of the problems (resistance alleles, carryover) that are the topic of the figure. it takes a bit of thinking to understand the significance of the fact that they are not just looking at progeny of heterozygotes, but progeny of heterozygotes that came from a specific sex. This would be a nice place to illustrate the things that go wrong and how they depend on grandparents. This has not really been presented schematically before, I think. Finally, panel A does not seem to be noted in the legend.

2. The paragraph lines 222-233 is a central part of the story. it would be useful to expand some of the single sentences into a few sentences to explain the results. For example, what is the thinking on the reduced fecundity of nos-expressing males. Is this likely to be male carryover, or is it just that perhaps the expression of Cas9 in males in the nos expression pattern is a bit toxic somehow to sperm? It would also be useful to unpack a little bit the key sentence that describes the zpg results, in terms of what it tells us.

"For zpg-CRISPRh we observed no reduction in male fecundity and the reduction in female fecundity was not subject to any maternal effect, consistent with the absence of maternal effect on homing for the same construct".

3. Figure 3 describes a truly elegant experiment designed to look at the effects of parental nuclease deposition, by following the fate of an existing R1 allele when in trans to a wildtype allele, and in the presence of Cas9 provided by one parental gamete type or the other through deep sequencing. This experiment also allows them to look for the appearance of new alleles due to Cas9 deposition..

The left panel, which schematizes the experiment, could be made much bigger and more bold. Also the colors for the different alleles are a bit too similar. it would be useful to make them more different and have a legend in the figure itself that makes clear which is which.

In the legend in part A there is no discussion of the F3 generation.

4. In Figure 4 the authors look at the what happens when homing does not occur. Are new alleles created, and if so what are they? There are a couple of things I am unclear about. First, in the cross scheme they have GD/DL twice in the F2, at 47%. Should one of these be GD/WT? Also, they do not explain in the text or the legend the significance of the docking line. Is this just an insertion at the target site that eliminates the target site and provides a molecular difference easily scored by sequencing and/or visually? In short, it would be useful to explain the screen a bit more. The text is a bit telegraphic as to what is being tested and how. Since this is an online journal I do not think there are strict length limits that require minimal descriptions.

Finally, it is interesting that most of the alleles rated in the female germline are R1. Is there something to speculate on here? Is it just that this specific site tends to mutate in a way that creates a specific R1, or might something else be going on?

5. In Figure 5 the authors report on the behavior of the zpg drive targeting nudel. We know it is going to fail, but the question is, given what has been demonstrated in the earlier figures, how much better will it behave, and what do the alleles that stop it look like.

First, it represents a lot of work to show when and what the resistance alleles look like, at many different time points. This figure stands alone in the gene drive field in this respect.

The authors discuss the egg counts in panel A, but dont refer back to a specific figure panel as showing this.

In panel B it would be useful to explicitly label which experiment is which since there are six panels with two Y axes labels (which is hard to see because they are so small).

it is unclear from their discussion of panel B what the data in the two sets of Y axes panels tells us, because they re not referred to. It is not that clear from the discussion what we take away from the figures versus the discussion, which focuses mostly on the number of mutant alleles that arise and how these are lower with zpg.

6. I also have a question that arises from the authors discussion. In the beginning of this section they say that "For the zpg-CRISPRh constructs, which have minimal fitness costs on fecundity in carrier females (containing one drive allele and one wild type allele), resistance did not show obvious signs of selection"

and then later on in the same section they say "The very late onset of resistance - if it is to occur - is a feature, not immediately intuitive, of this type of gene drive whose goal is suppression and where heterozygous ‘carriers’ are intended to show minimal fitness costs. Selection for r1 alleles only really becomes strong when the gene drive is close to fixation - when most individuals have at least one copy of the gene drive the relative advantage of an r1 allele over remaining WT alleles (that have a high probability of being removed by homing of the gene drive"

I appreciate what they are trying to say in the second paragraph. A key feature of a suppression HEG is that so long as heterozygotes are mating with wildtypes no costs are incurred–for an IDEAL HEG. However, in the earlier data, Fig. 2, they show that zpg females suffer a 50% fecundity cost. While this is much lower than the costs associated with vasa, I think most people would still consider a 50% fecundity cost to be quite large. It would be useful to rethink the wording for the first statement, and link these costs to the second statement. While in ideal world this is how a HEG should behave, how does a 50% fecundity cost alter this calculation. Also, is it correct to say that resistance evolves late, or just that it manifests its effects late in the drive process, at least when only ideal HEG behavior is considered.

In short, it might be useful to reframe this section a bit, taking into account the fecundity costs, when R1 alleles arise, etc. I appreciate that the end result is clear in terms of increased drive persistence, but the arguments for how the observed behavior gets us there might be more clear. Again, I dont see space as a limitation.

7. In figure 6, panel lower left, it looks like the blue line goes to zero and then undergoes a dramatic increase. Can you explain?

Also, it looks like the data points are collected every other generation, 2-4-6. In this context the three tick marks between each set of numbers are a bit confusing and could be eliminated (since they cannot mean anything biological).

8. In the conclusions they say

"Indeed when the zpg promoter was used to control expression of a homing based gene drive at the doublesex locus we noticed a paternal effect contributing to reduced fecundity that was not observed here [8]" Can the authors expand for a sentence or two on how this is likely to occur: cas9 expression having effects on sperm fitness or paternal carryover? or something else. It would just be good to have a clear understanding of possible sources, since even some non-homing clvr-type drives proposed for suppression require germline only effects, and Cas9 is generally considered not to show paternal carryover (though most researchers do not look carefully for it).

////////////////////////

Very minor points.

1. We tracked the generation and selection of resistant mutations the course of a gene

It looks like the authors missed a word, perhaps "over" the course"

2. onset of target site resistance. Our results show that regulation of its expression

Maybe just change the word it to nuclease, to make clear what is being referred to

3. developing a robust gene drive.

...for homing based suppression

4. throughout the text the authors use the standard error of the mean, rather than standard deviation. This is just my personal opinion, but I find SEM values completely useless. It actually hides the diversity of outcomes that would be apparent on simple visual inspection of the data (like in figure 2). I like SD because it actually tells you something about the distribution. But, its just a suggestion.

5. In the methods zpg-crispr is listed twice at line 488

6. is this the docking line referred to in the text? If so it would be useful to expand a bit in the text as to the gfp-based method used here to identify these individuals.

"500 female zpg-CRISPRh drive heterozygotes (F0) were crossed to a line heterozygous for a GFP+ marked knock-out AGAP007280 allele and allowed to lay eggs. Upon egg hatching, L1 larvae were sorted using COPAS (complex object parametric analyser and sorter), as in Marois et al. (2012). The very rare fraction of progeny (F1) that inherited the GFP+ marked knockout but not the zpg-CRISPRh allele due to lack of homing..."

7. The authors refer to references 15 and 20 for data showing that zpg RNA is localized.

Line 123 " deposited nos and zpg mRNA concentrate at the germ plasm due to regulatory signals present on the untranslated regions, which also further restrict translation of maternal mRNAs to the germline [15, 20]." I have looked at these references and do not see any evidence that zpg mRNA is localized to germ plasm. Is this correct? it is of course for nanos, but I am not familiar with any localization data for zpg, particularly since it acts much earlier during oogenesis.

8. In supplement figure 2 legend the authors state that " (A) Increasing the fertility of heterozygous zpg-CRISPRh females by up to 80% can accurately predict the experimental data, suggesting the fertility assays underestimate their true fecundity."

Can they explain this a bit more and relate it to the data presented in Figure 2 on fecundity" What exactly do they mean here and how does it relate to Figure 2? More generally, should some or all of these observations be moved to the main text given that they seem to offer a set of discussion points (in the legend) that are distinct from those made in the text regarding how one should think about the data presented in the text. In short, it is confusing to present data in the text and then say in the supplement we get a better fit when we increase fecundity over what is obsrved, therefore the data in the text must not accurately reflect the real situation.

9. In supplementary Figure 4 it is not clear what the green highlights of specific bases mean. Are these sequence polymorphisms from the reference genome, or something else?

Reviewer #2: The manuscript "Regulating the expression of gene drives is key to increasing their invasive potential

and the mitigation of resistance" explores and discusses the role of different regulatory sequences on the potential for Cas9 gene drives to generate resistance alleles which would break said drives.

The authors start by outlining the known issues around generating functional gene drives, and the scenarios via which resistance may develop and be selected for. They then detail the development of three new gene drive strains targeting a single locus associated with female fertility, and compare this to their previously published gene drive targeting the same locus.

This is a very interesting topic, and builds nicely on the authors previous work. Overall this a very nicely written and designed set of experiments, with only a few minor comments.

See below my comments ordered by section.

Abstract: I found the reference to modifications to germline regulatory sequences slightly misleading. This work compared several endogenous regulatory sequences, but didn't start editing or modifying these sequences. Could the authors clarify this statement

Introduction:

This section is well written and lays out the rest of the MS clearly

L72 - this sentence doesn't make sense grammatically

Results:

Data analysis - The use of M-W tests and KW tests is fine - though the authors could consider whether using a linear model would eliminate the small risk of inflated type 1 error rates from running mutliple comparisons. In addition - a G-test/Chi-square approach to rates of inheritance would allow the calculation of odds ratios between lines e.g. males inherit at X rate compare to females - and direct rather than indirect comparisons between lines.

I would also caution that the use of these nonparametric tests indicates that the data has a non-normal distribution, in which case the use of means and s.e. should be avoided and replaced with median and IQR instead.

It would also be good to see the full statistics includede here e.g. test statistics and sample sizes, not just the p value

In Figure 2 - I struggled to compare these back to the numbers listed in text - are these different replicates that have been put together for analysis, if so this needs to be made explicit. In addition details in the legends should make it clear what points, lines and error bars represent.

L211 - It would have been nice to see egg laying data as well as larval output, to look at egg laying and viability

L253 - Brown-Forsythe and Welche are two different types of ANOVA - so which one was used?

Figure 4 - refers to R2 as non-resistant mutations and should be corrected

L316 - elaborated on later in the discussion but worth emphasising - that efficient drives can be release at low numbers and be as effective (if not as quick) at reaching high frequency - is a powerful finding

L329 - the pooling method could be more explicit - rather than refer to another MS briefly outline sample types and numbers

L362 - In the creation of resistant mutants there is not much discussion of the role of mosaicism in the vas2 results. Why do you think you get more r2 alleles? Is it because the increased rates of mosaicism mean you detect these in individuals that also have uncut functional alleles?

L370 - You outline a readily testable idea - that these different alleles results are stochastic and not based on selection - even without performing the experiment, it would be nice to see a brief outline of how you would confirm that

L374 - speculation on the idea that there are fewer r1 allele variants in the 'new drive' there could also be other explanations? Lakc of mosaicism in these lines means fewer alleles are able to restore fertility to 100%? Or that timing of ediitng plays a role in the rate of NHEJ, so fewer indels generated?

Reviewer #3: In this study, the authors assess the performance of several promoters for Anopheles gambiae Cas9-based gene drives. They find that the Zpg promoter has superior performance compared to the Nos and especially the Exu promoters in terms of drive inheritance. The Zpg promoter also outperforms the previously utilized Vas2 promoter by reducing somatic expression, thus restoring partial fertility to females with the gene drive. Overall, this is a useful paper that provides new insights into gene drive mechanisms. The study is technically sound, assuming that the comments raised below can be addressed.

Some of the results presented in this paper have already been described in a paper published by the same group in Nature Biotechnology in 2018. However, an earlier version of the current paper was available as a preprint (https://www.biorxiv.org/content/10.1101/360339v1?versioned=true) at that time, and this preprint is cited in the Nature Biotechnology paper.

Specific comments:

1. Within each construct, did the individuals with lower fertility also have lower drive inheritance among their offspring? Exu seems to have a different horizontal axis scale than the other four panels of Figure 2, which the authors might want to adjust for ease of comparison. Why is there no statistically significant difference between the Nos male drive inheritance between paternal and maternal drive alleles? Can this be double-checked? It looks like a fairly sizable difference.

2. Why was the wild-type fecundity different for each promoter, when these insects should have been free of the transgene? Were they part of the same batch as the transgenic mosquitoes of each promoter, or were they all independent sets of experiments? This should be noted. If the latter, this possibly indicates confounding batch effects when assessing the relative fecundity of the different transgenic lines. In this case, probably only the larger differences should be considered statistically significant, or an alternative analysis should be performed that takes batch effects into account using all wild-type crosses for each comparison.

3. Why was male fecundity reduced for nos and exu (but not vasa and zpg) if the target gene only affects female fertility? Could this be a line effect (the line of transgenics simply being less healthy in some cases due to the transgenesis itself, rather than the specific transgenes) or a batch effect as described above?

4. Why was fecundity not determined for the vasa males with drive mothers? Why did these males have slightly biased inheritance toward the gene drive if they were heterozygous for a resistance allele? Also, “ND” should be defined in the legend.

5. The cage data seems to deviate from the model simulations, especially for eggs laid. This could be due to a model parameter mismatch, but a presumably more likely explanation would be that the effective population size in the cages was lower than the census size, yet the latter was used in the model. It would therefore be interesting to adjust the model to a lower population size if possible. This issue of effective population size should at least be noted in the results.

6. Lines 266-271: in this experiment, the authors did not specifically select individuals which did not develop resistance alleles in the embryo to test for shadow drive. Since resistance allele formation in the embryos was near total for vasa, any “signal” for shadow drive would be minimal. For the other promoters, lack of any resistance allele formation in the embryo indicates that there is little to no maternal deposition, which would also rule out the possibility of shadow drive (if no Cas9 at early stages, none would be available later in the germline). At best, the authors can probably rule out “extremely strong” shadow drive (note that the Drosophila studies found at most moderate shadow drive).

7. Lines 292-294: early germline resistance alleles were also seen in references 10, 13, and 32, which could be cited here as well.

8. The references on lines 423-424 might not be the most appropriate for showing that multiplexing improves the robustness of drives. Reference 21 didn’t involve using multiple gRNAs to improve the efficiency of a drive (the gRNA targeted other sites that were not involved in the main drive), and reference 37 was a different form of gene drive and thus not fully relevant to the question of multiplexed gRNAs in homing drives. Reference 12 would be good here, as would this new paper: https://www.pnas.org/content/early/2020/09/11/2004373117

In line 425, reference 37 did successfully reduce r1 resistance alleles, possibly through targeting constrained sites, but the above PNAS reference would also be a suitable demonstration of this. Really, though, only reference 8 conclusively shows the importance of constrained target sites.

9. The authors should provide their Mathematica modeling code as a supplement or in a permanent online repository such as GitHub.

10. A new paper has recently used the nos promoter in Anopheles and should probably be cited: https://www.pnas.org/content/117/37/22805

**Have all data underlying the figures and results presented in the manuscript been provided?**

Reviewer #1: Yes

Reviewer #2: Yes

Reviewer #3: Yes

PLOS authors have the option to publish the peer review history of their article (what does this mean?). If published, this will include your full peer review and any attached files.

Reviewer #1: **Yes: **Bruce A. Hay

Reviewer #2: No

Reviewer #3: No

---

## [Decision Letter · Decision Letter 1]

5 Dec 2020

Dear Dr Nolan,

Thank you very much for submitting your Research Article entitled 'Regulating the expression of gene drives is key to increasing their invasive potential and the mitigation of resistance' to PLOS Genetics.

The manuscript was fully evaluated at the editorial level and by independent peer reviewers. The reviewers appreciated the attention to an important topic but identified some concerns that we ask you address in a revised manuscript

We therefore ask you to modify the manuscript according to the review recommendations. Your revisions should address the specific points made by each reviewer.

[LINK]

Yours sincerely,

Owain Rhys Edwards, Ph.D.

Guest Editor

PLOS Genetics

Gregory Barsh

Editor-in-Chief

PLOS Genetics

Reviewer's Responses to Questions

**Comments to the Authors:**

Reviewer #1: All of my concerns have been addressed adequately.

Reviewer #2: This is my second review of The manuscript "Regulating the expression of gene drives is key to increasing their invasive

potential and the mitigation of resistance" I am pleased to see that my original queries have all been addressed, and in my opinion satisfactory responses to the other reviewer comments as well.

This is a welcome new piece of research to the field of gene drive development and I will be happy to see it published

Reviewer #3: The authors have addressed some of my points, but there are still some remaining issues I feel should be addressed before publication.

1. Differences in inheritance between paternal and maternal alleles in the Nos male drive

The authors note that they double checked and found that there are indeed no significant differences. This seems strange, so I ran a Mann-Whitney test myself. Indeed, it was not significant (p=0.1). However, the difference between the female lines (with either male or female drive parents) only had p=0.03 instead of p<0.001 as shown in the figure. I’d therefore suggest rechecking these statistics.

2. Difference in wild-type fecundity for the different promoters:

The authors state in their response that “Given variability in bloodmeal batch/insectary conditions etc we find it most robust to have a matched wild type control. We have noted this in the methods as per the reviewers suggestion”.

This is good, but keep in mind that the controls did indeed have different performance. The authors cannot therefore assume that each control batch is suitable for comparison. The differences between experiments could well have been accounted for by their paired control, but it’s also possible that there is an individual batch effect that can separately affect each batch within the paired experiments. For example, the wild-type control in the vasa batch may have had worse than “usual” fecundity, but the vasa samples could have been fine. The wild-type samples in the Exu experiments could have had better than average fecundity, but that doesn’t mean that the Exu samples within this experiment themselves were also having a good day, thus indicating that the Exu mosquitoes have reduced fecundity. Instead, they could be performing average, with Exu not reducing fecundity at all.

Thus, some of the conclusions about fecundity being reduced in some lines should probably be softened in the results section, and the possibility of these batch effects noted there in the results, not just the methods. Looking at this, I suspect that the Exu mosquitoes might not actually have reduced fecundity, and even the Zpg female reduction and the nos reduction are somewhat questionable (though based on the zpg/dsx paper, the Zpg fecundity reduction is likely quite reliable).

3. Reduced fecundity in nos-CRISPRh males

I am glad to see that the authors now discuss the anomalous reduction in male fecundity. However, I would encourage mention of the above point - that the reduction in fecundity could just be due to batch effects of the experiment and not an actual feature of the drive.

4. Deviations between cage data and model simulations.

I’m generally happy with how the authors revised this. However, I feel they should also mention that variation in male mating success has a large impact on the level of stochastic variation in the cages.

5. Shadow drive.

The authors state in their reply that “While this experiment was not looking for vasa-driven shadow drive of a gene drive allele it was designed with the sensitivity to pick up vasa-driven shadow drive of the r1 resistant allele since it was balanced against a wild type cleavable allele.”

Yes, I understand this. However, shouldn’t the r1 allele be able to do shadow drive only if a wild-type allele is available for conversion? It looks like most wild-type alleles were converted to resistance alleles in the early embryo in this experiment (only about 5% appear to have remained wild-type), hence the reduction in power because they would not be available for conversion in the germline. I thus don’t think the authors can generally rule out shadow drive at the level of the Drosophila studies, even though very high levels of shadow drive can indeed be ruled out.

**Have all data underlying the figures and results presented in the manuscript been provided?**

Reviewer #1: Yes

Reviewer #2: Yes

Reviewer #3: Yes

PLOS authors have the option to publish the peer review history of their article (what does this mean?). If published, this will include your full peer review and any attached files.

Reviewer #1: **Yes: **Bruce A Hay

Reviewer #2: No

Reviewer #3: No

---

## [Editor Report · Decision Letter 2]

22 Dec 2020

Dear Dr Nolan,

We are pleased to inform you that your manuscript entitled "Regulating the expression of gene drives is key to increasing their invasive potential and the mitigation of resistance" has been editorially accepted for publication in PLOS Genetics. Congratulations!

Yours sincerely,

Owain Rhys Edwards, Ph.D.

Guest Editor

PLOS Genetics

Gregory Barsh

Editor-in-Chief

PLOS Genetics

Comments from the reviewers (if applicable):

**Data Deposition**

http://datadryad.org/submit?journalID=pgenetics&manu=PGENETICS-D-20-01181R2

**Press Queries**

---

## [Editor Report · Acceptance letter]

23 Jan 2021

PGENETICS-D-20-01181R2 

Regulating the expression of gene drives is key to increasing their invasive potential and the mitigation of resistance 

Dear Dr Nolan, 

We are pleased to inform you that your manuscript entitled "Regulating the expression of gene drives is key to increasing their invasive potential and the mitigation of resistance" has been formally accepted for publication in PLOS Genetics! Your manuscript is now with our production department and you will be notified of the publication date in due course.

With kind regards,

Alice Ellingham

PLOS Genetics

On behalf of:
